# Association of adult caregiver depression with developmental disorder likelihood in Ugandan children perinatally exposed and unexposed to HIV

## Research Article

depression; caregivers; children; disorders; assessment

**Corresponding author:**
Jorem Emmillian Awadu;
Email: awadujor@msu.edu

Jorem Emmillian Awadu[1] , Bruno J. Giordani[2], Alla Sikorskii[1], Sarah Zalwango[3], Catherine Abbo[4] and Amara Ezeamama[1]

[1]Department of Psychiatry, College of Osteopathic Medicine, Michigan State University, East Lansing, MI, USA; [2]Psychiatry, University of Michigan, Ann Arbor, MI, USA; [3]Public Health, Kampala Capital City Authority, Kampala, Uganda and [4]Psychiatry, Makerere University CHS, Kampala, Uganda

## Abstract

We assessed whether higher caregiver depression is associated with increased likelihood of caregivers rating their children as screening positive for developmental disorders—autism spectrum disorder (ASD), attention-deficit/hyperactivity disorder, emotional behavioral disorder, and functional impairment (FI)—among Ugandan children perinatally exposed and unexposed to HIV. Children and their primary caregivers were followed for 12 months. Caregiver depression was measured using the Hopkins Symptom Checklist-25 and categorized as low, moderate, or high based on terciles. Child developmental indices were derived from the Behavioral Assessment System for Children (third edition) at 0, 6, and 12 months. Multivariable linear regression estimated mean differences (MDs) in disorder indices with 95% confidence intervals (CIs) by caregiver depression level. Compared with highly depressed caregivers, those with low depression reported consistently lower ASD risk scores (MD: −0.35 to −0.32; 95% CI: −0.60 to −0.08). Similar trends were observed for FI (MD: −0.56 to −0.31; 95% CI: −0.81 to −0.06). Moderate depression was associated with modestly lower FI risk at baseline and 6 months but not at 12 months. Overall, higher caregiver depressive symptoms were linked to greater perceived child disorder risk. Evaluating caregiver depression alongside child screening may improve interpretation of developmental risk assessments.

## Impact statement

Depression is a highly prevalent comorbidity among caregivers of dependent children with chronic conditions. It is known to interfere with one's functioning and yet is high in low- and medium-income countries (LMICs), especially among females who provide most of the childcare, including but not limited to, hospital visits for children's medical care. Understanding how depression in the caregiver impacts their subjective reporting of neurodevelopmental disorder symptoms in their dependent children is important for early and accurate diagnosis and timely linkage to available resources.

This investigation among adult caregivers from the LMIC setting of Uganda found that low caregiver depression was associated with underreporting of neurodevelopmental disorder symptoms in their respective dependent children.

We highlight the need to consider caregiver depression in the contextual understanding of their report of symptoms in dependent children in order to optimize diagnostic accuracy and to efficiently utilize limited resources. Whenever possible, multidimensional assessments of developmental disorders and behavioral outcomes should be used, including, as appropriate the child's, the caregiver's or other respondent's perspective.



## Introduction

Between 1990 and 2019, the prevalence of mental health disorders increased by 48.1%, rising from 654.8 million (95% uncertainty interval of [603.6–708.1] to 970.1 million (95% uncertainty interval of [900.9–1,044.4]) (Collaborators GBDMD, 2022). Depressive disorders affect over 280 million people globally, with the most burdened being females (approximately 170 million) and persons living in sub-Saharan Africa (Collaborators GBDMD, 2022). The prevalence of depressive disorders in the general population of people from sub-Saharan Africa is 4.3% (Collaborators GBDMD, 2022). Within Africa, depression is underreported (Lofgren et al., 2020). Depression is reported to be high in persons living with HIV (PLWH) (Bernard et al., 2017) and among African youth between 10 and 24-year-olds (Too et al., 2021). Specifically,

published reviews indicate an average depressive disorder prevalence rate of 25.5% among PLWH with estimated rates across studies ranging from a low of 13% to a high of 78%. These findings compare to a depressive disorder mean prevalence rate of 5% and lifetime depressive disorder risk of 15%–18% in the general population (Lofgren et al., 2020). In the global population of women living with HIV, the estimated prevalence of depression is even higher at 82% (Orza et al., 2015).

The etiology of depression is multifactorial, with a range of biological and contextual factors implicated. Regardless of the cause, depression is associated with poor functioning (Lin et al., 2024) parenting practices that are insensitive to a child's needs, and spousal difficulties, (Liskola et al., 2021; Wang et al., 2022; Lin et al., 2024), which negatively impact child development and behavior (Familiar et al., 2020). It also changes the threshold by which a caregiver determines their child's behavior needs attention with a possibility of under reaction or overreaction, hence impacting their own subjective interpretation of their child's behavior with consequence for parent report in neuropsychological tests. Indeed, published data suggest that depressed caregivers are more likely to overreport negative child behavior,(Familiar et al., 2020; I. Familiar et al., 2016). A similar relationship is reported for caregiver depression, child emotional and behavioral problems,(De Los Reyes and Kazdin, 2005; Hennigan et al., 2006; Gartstein et al., 2009) as well as externalizing and internalizing symptoms – facets central to clinical diagnostic decisions in attention deficit hyperactivity disorder (ADHD), autism spectrum disorder (ASD) and oppositional defiant disorder (ODD) (Ayano et al., 2019; Vizzini et al., 2019). Hence, caregiver depression screening is crucial for proper contextualization and interpretation of results from proxy-evaluated child cognition or behavioral outcomes. Also, dealing with developmental disorders (most often medically based) can cause stress on caregivers suggesting the need to evaluate caregivers for situational stress.

Developmental disorder prevalence rate is rising in the general population,(Zablotsky et al., 2015; Sharma et al., 2018; Collaborators GBDMD, 2022) including among children exposed and unexposed to HIV/antiretroviral therapy (Too et al., 2021; Awadu et al., 2022; Mpango et al., 2022). Whether this rise is due to improved screening, clinician/professionals training or higher overall awareness is still not well elucidated. However, there is relative consensus that some of the rising trend is attributable to increased awareness of developmental disorder symptomology among caregivers, child development and medical professionals (Elsabbagh et al., 2012). Unlike disorders with known biomarkers, the diagnosis of most developmental disorders relies upon psychological assessments, observation and information gathering from caregivers knowledgeable of the child to be assessed (Müller et al., 2011; Liskola et al., 2021). Caregivers are often asked about the developmental history and behavioral presentations of the child of interest. Such requests usually require complex cognitive functioning to ensure the accuracy of recall of the child's developmental milieu and history, as well as of the frequency of common symptom patterns. It is common for child behavioral ratings by two or more primary caregivers of the same child to have discrepancies (De Los Reyes and Kazdin, 2005; Müller et al., 2011; Liskola et al., 2021). Inaccurate child ratings lead to misclassification or misdiagnosis, can strain limited available public health or personal resources, and lead to missed intervention opportunities at crucial developmental time points. One factor leading to inaccurate child ratings by caregivers has been characterized as the depression-distortion hypothesis – which posits that caregiver depression negatively biases their evaluation of behavioral

or emotional problems in their children (Müller et al., 2011; Liskola et al., 2021).

The influence of maternal depression on maternal ratings for child behavior has been reported mostly in high-resource countries (Collaborators GBDMD, 2022). However, Familiar and colleagues (Familiar et al., 2020; I. Familiar et al., 2016) found that, regardless of HIV status, depression symptoms among caregivers in Sub-Saharan Africa were associated with their reports of child behavioral problems. However, the relationship of caregiver depression to caregiver-reported developmental disorder risk in vulnerable dependent children from low- and middle-income countries (LMICs) has not been investigated. Understanding this relationship in LMICs like Uganda with a high HIV burden, limited mental health infrastructure, inadequate number of trained clinicians and rehabilitation professionals is necessary to ensure limited resources are focused on children with the highest need. This study seeks to better characterize the influence of maternal depression on caregiver reporting of developmental disorder symptoms by investigating whether depression is associated with the caregiver's rating of developmental disorder likelihood risk scores for their dependent HIV-exposed and unexposed children. It was hypothesized that caregiver depression level would be associated with higher risk perception and thus higher developmental disorder risk scores in their dependent children. Results obtained from this study (i.e., a secondary analysis of a prospective cohort) will enhance our understanding of the role of depression in developmental disorder ratings by caregivers – a factor that can be considered and potentially adjusted for during their assessment.

## Methods

### Study participants, study setting, recruitment and follow-up

Six hundred and three children of known perinatal HIV status at 6–18 years old together with their current adult (i.e., ≥18 years old) caregivers were recruited and followed for 12 months. G*Power (Faul et al., 2009; Faul et al., 2007) was used to determine that a sample size of 387, with 95% power for a two-sided significance test at an alpha level of 0.05, would allow us to detect the small to moderate effect size differences observed in similar studies conducted in Uganda (Itziar Familiar et al., 2016). Our sample is thus adequate to identify small differences between the study groups. Included were children perinatally infected with HIV (CPHIV, n = 203), children HIV/ART exposed but uninfected (CHEU, n = 198) and children HIV unexposed and uninfected (CHUU, n = 202). Caregivers of dependent children were recruited regardless of their current HIV status. Caregivers and dependent children were recruited through two cohort studies conducted in Kawaala Health Center IV (KHC), Kampala, Uganda, between 16 March 2017 and 30 June 2021. CPHIV and their adult caregivers were primarily recruited from those already receiving primary care at KHC clinics. CHEU and CHUU were recruited concurrently with CPHIV in three ways: (a) for CHEU, we leveraged the early infant diagnosis system and invited caregivers of HIV-exposed children whose children were born in the labor and delivery unit of KHC; (b) we encouraged co-enrolment of CHEU and CHUU within the same households as already enrolled CPHIV and (c) we leveraged the social networks of already enrolled families in the project to invite CHUU and CHEU directly from the community. HIV-negative status at enrolment was confirmed for CHEU and CHUU using HIV rapid diagnostic test. We implemented a secondary analysis of data collected as part of a prospective cohort study of

caregiver depression level in relationship to their subjective rating of their dependent child's developmental disorder probability risk score in Kawaala Health Center (KHC), Kampala, Uganda. By design, an equal proportion of children were recruited into the different study arms. Study participants were followed for 12 months with developmental disorder scores assessed at enrollment, 6, and 12 months or until lost to follow up.

### Study eligibility/exclusion criteria

All study children not born in a health facility were excluded (for both cohort studies) because HIV status for them and their biological mothers, their ART exposure in utero/peripartum, as well as HIV status for both the index child and their biological mother could not be objectively determined based on tests done as part of antenatal care or the PMTCT program. All eligible study caregivers and their respective children had to live within 25 km of the study area (i.e., KHC in Kawempe Division, Kampala, Uganda) with no known plans to relocate outside the study area at enrolment. Additionally, for this secondary analysis, children without developmental disorder risk or caregiver depression information were excluded.

### Ethical approval

Institutional review boards from Makerere University (Protocol REC REF numbers: 2017-017 and 2018-099) and Michigan State University (IRB Protocol numbers: 16–828 and 205), respectively reviewed and approved the study protocol and all study forms before the studies commenced. Additionally, the Uganda National Council for Science and Technology (Protocol numbers: SS4378 and HS 2466) reviewed and provided the final permission for the study to start. Adult caregivers provided informed consent, and their dependent children provided assent for participation in the study.

### Outcomes: Developmental disorder, resiliency and functional impairment probability scores

Probability risk score for ASD, ADHD, emotional behavioral disorder (EBD), functional impairment (FI) and resiliency indices (RI) were derived per criteria provided in the Behavioral Assessment System for Children, Third edition (BASC-3) manual (Zhou et al., 2022) and reported in our previous publications (Ezeamama et al., 2021). Per standard protocol,(Zhou et al., 2022) questionnaires were administered in the respondents' language. BASC-3 items were forward translated to the local language of Luganda and then back translated to English with an expert panel to adjudicate disagreements. Further, questions were culturally adapted for the research context with care taken to ensure item meaning was culturally relevant while preserving intended meaning of the original tool as we have described elsewhere (Ezeamama et al., 2021). Scores for each disorder were age and sex standardized to the mean and standard deviation of baseline scores of apparently healthy children without perinatally acquired HIV. The resulting age and sex standardized z-scores were analyzed as response variables. This approach assures a contextually relevant reference group for interpretation of neurodevelopmental outcomes as reported in our prior studies (Ezeamama et al., 2021).

Primary Predictor: Caregiver Depression. At baseline, caregiver depression level was measured using 15 items in the depression subscale of the Hopkins Symptom Checklist-25 (HSCL-25). Participant response to each questionnaire item was scored on the Likert scale from 1 (not at all) to 4 (often). Ratings were summed, and then low (0–9), moderate (10–15) versus high (≥16) categories of caregiver depression level were defined based on the tercile of total scores for analytic purposes. We used sample specific cut-offs as best operational fit for definition of relative caregiver depression for two reasons. First, although the HSCL-25 is a validated tool for depression, that validation was not in sample directly comparable to our study population. Second, our goal was to make relative comparisons of caregivers' perception/assessment of their child's outcomes according to relative severity of depressive symptoms. Thus, using empiric distribution provided maximum power in respective categories with fidelity to our study question.

### Other measures

Sociodemographic variables: Child and caregiver age (in years) and biological sex (male vs. female) were recorded at baseline. HIV status at birth was derived from hospital records (i.e., antenatal registers or notes, labor and delivery forms, antiretroviral cards) at KHC. Child current HIV status at enrolment was ascertained using HIV rapid diagnostic test.

Caregiver education, socioeconomic and lifestyle factors: Caregiver education was defined as years of formal education. The presence or absence of income source was used as a proxy indicator of socioeconomic status. Smoking and alcohol use history were defined as ever versus never. Information on life adversity and living with sexual partner was gathered using a structured questionnaire (Brugha et al., 1985; Glover et al., 2010; Goodman et al., 1998). Current caregiver HIV-status was determined by their response to a specific question about current HIV status at the baseline assessment.

### Statistical analyses

As part of the descriptive analysis, analysis of variance (ANOVA) and chi-square tests were used to compare continuous and categorical variables according to respective caregiver depression symptom levels. Thereafter, multivariable repeated-measures linear regression models using quantified caregiver mean differences (MDs) along with corresponding 95% confidence intervals (95% CI) in age- and sex-standardized outcome measures according to categories of baseline caregiver depression using SAS PROC MIXED. Potential confounders, like child HIV status; time categories (categories: 0, 6 and 12 months); caregiver sex; age; education; lifetime adverse experiences and lifestyle factors (smoking and alcohol use), were adjusted for based on subject-matter knowledge.

We examined the potential for differences in depression association with respective outcomes over time by including a time-by-depression interaction. Associations for depression with respective outcomes were presented separately within follow-up intervals when the p-value for the depression-by-time interaction was <0.1. When the p-value for the depression-by-time interaction was >0.10, time-averaged association of depression with respective outcomes were presented. In all analyses, we accounted for lack of independence among children living in the same household by including household ID as random effect. Given that disorder outcomes were standardized by age, estimated MDs have similar interpretations as Cohen's d effect sizes, thus providing insight on the clinical importance of depression-associated differences in outcome measures. Small, moderate and large levels of clinical importance were deduced basing on

MD thresholds of <0.33, 0.33 ≤ |MD| < 0.50 and |MD| ≥ 0.50, respectively. (Baron-Cohen et al., 2001) Specifically, SPSS version 25 (Landau and Everitt, 2003) was used to derive BASC-3 and HSCL-25 reliabilities (Cronbach's alpha) while all other analyses were performed in SAS version 9.4 (SAS Institute, Inc., Cary, NC, USA). All hypothesis tests utilized were two-sided at alpha = 0.05.

## Results

Of the 603 children (whose data were completed by the time of this analysis) assessed for this study, 143 (23.7%), 305 (50.6%) and 155 (25.7%) had caregivers who were in the low-, moderate- and high-depressed symptomology subcategories, respectively, per the HSCL-25. The average age in years for the study children was 11.34 (SD =3.65) and the majority were female 315 (52.24%). Children in respective caregiver depression symptomology levels were significantly different for age. Dependent children of caregivers with high depression symptomology were slightly younger (M = 10.69, SD = 3.78, years old) and mostly CHEU. Study caregivers were of similar age with an overall mean age of 38.96 (SD = 11.64) years as well as years of education. However, caregivers had a significant difference in the number of lifetime adverse experiences with

caregivers in the high depression symptomology category having experienced the most adversity (i.e., M = 3.42, SD = 2.83). Most caregivers were female (88.49%), living with HIV, had own source of income and a history of alcohol use. (Table 1).

Overall, regardless of depression symptom level, caregivers' rating of their children for developmental disorders (i.e., ASD, ADHD and EBD) and FI risk indices improved significantly by 1 year of the study. Caregivers with low depression symptoms rated their children as having a moderate lower risk for all developmental disorders and FI outcomes (all p < 0.05) at 12 months. The same temporal trend, that is, an improvement in children's probability risk measures, was observed over time regardless of caregiver depression symptom level. (Supplementary Table S1).

The BASC-3 showed poor internal consistency, with Cronbach's alpha values of 0.55 for both ASD and ADHD probability indices. However, it demonstrated good reliability with 0.72 and 0.75 for EBD and FI probability indices, respectively, and an acceptable reliability of 0.62 for RI. In contrast, the HSCL-25 exhibited good reliability for both caregiver depression and anxiety with Cronbach's alpha values of 0.87 and 0.82, respectively (Table 2).

Adjusted for time, HIV status, sex and age (in children) and HIV status, sex, age, education, lifetime adverse experiences and history of smoking and alcohol use (for caregivers), the caregivers with low

**Table 1.** Baseline description of children and adult caregiver pairs from Kampala Uganda with respect adult caregiver depression levels

| | Overall (N = 603) | Low (n = 143) | Moderate (n = 305) | High (n = 155) | One-way ANOVA (continuous variables)/Chi-square test (categorical variables) |
|---|---|---|---|---|---|
| **Child factors** | | | | | |
| | **Mean (SD)** | **Mean (SD)** | **Mean (SD)** | **Mean (SD)** | |
| Age | 11.34 (3.65) | 12.33 (3.48) | 11.22 (3.57) | 10.69 (3.78) | 0.0003 |
| | **n (%)** | **n (%)** | **n (%)** | **n (%)** | |
| **Female sex** | 315 (52.24) | 76 (53.15) | 153 (50.16) | 86 (55.48) | 0.543 |
| **Perinatal HIV status** | | | | | 0.033 |
| CPHIV | 203 (33.67) | 64 (44.76) | 98 (32.13) | 41 (26.45) | |
| CHEU | 198 (32.84) | 37 (25.87) | 90 (29.51) | 71 (45.81) | |
| CHUU | 202 (33.50) | 42 (29.37) | 117 (38.36) | 43 (27.74) | |
| **Caregiver factors** | **Mean (SD)** | **Mean (SD)** | | **Mean (SD)** | |
| Age | 38.96 (11.64) | 40.97 (13.22) | 38.05 (11.05) | 38.88 (11.08) | 0.145 |
| **Education level** | 3.76 (3.02) | 3.84 (3.18) | 3.75 (2.95) | 3.67 (3.03) | 0.892 |
| Lifetime adverse experiences | 2.21 (2.36) | 1.02 (1.23) | 2.13 (2.20) | 3.42 (2.83) | **<.0001** |
| | **n (%)** | **n (%)** | | **n (%)** | |
| **Female sex** | 523 (88.49) | 110 (79.71) | 276 (92.00) | 137 (89.54) | **0.0008** |
| **Living with HIV** | | | | | |
| *Yes* | 381 (63.61) | 89 (62.24) | 184 (60.78) | 108 (70.59) | 0.109 |
| **Socioeconomic status** | | | | | |
| *Has own income source (Yes)* | 452 (75.21) | 110 (76.92) | 222 (73.03) | 120 (77.92) | 0.448 |
| **Living with a sexual partner** | 277 (45.94) | 70 (48.95) | 135 (44.26) | 72 (46.45) | **0.643** |
| **Behavioral factors** | | | | | |
| Ever smoker | 44 (7.30) | 18 (12.59) | 10 (3.28) | 16 (10.32) | **0.001** |
| Ever used alcohol | 261 (43.28) | 67 (46.85) | 112 (36.72) | 82 (52.90) | **0.003** |

*Note:* Unadjusted comparisons.

levels of depression reported moderately lower ADHD (MD: -0.36; 95% CI: −0.57, −0.15) and EBD (MD: -0.57; 95% CI: −0.78, −0.36) probability scores for their dependent children relative to caregivers with high levels of depressive symptoms. Similarly, caregivers with moderate depressive symptom levels reported lower probability scores for the same outcomes in their dependent children relative to caregivers with high depression. This difference amounted to moderately lower EBD probability scores for dependent children of caregivers with moderate depression level only. We found no evidence that the relationship of caregiver depression to respective outcomes varied over the follow-up period (time-by-depression interaction, p > 0.10); hence, time-averaged associations are shown (Figure 1, Supplementary Table S2).

The relationship between caregiver depression symptomology, ASD and FI varied over time. Caregivers with low levels of depression symptoms rated their dependent children as having a low-to-moderate statistically significant likelihood for ASD at baseline

**Table 2.** Reliability of the Behavioral Assessment System for Children (BASC) and Hopkins Symptom Checklist-25 (HSCL-25) in children from Uganda

| | N | Number of items | Internal consistency (Cronbach's alpha) |
|---|---|---|---|
| **BASC–3 subscales** | | | |
| Attention deficit hyperactivity disorder probability index | 27 | 9 | 0.55 |
| Autism spectrum disorder probability index | 27 | 18 | 0.55 |
| Emotional behavioral disorder probability index | 27 | 20 | 0.72 |
| Functional impairment index | 25 | 41 | 0.75 |
| Resiliency | 27 | 13 | 0.62 |
| **Hopkin's Symptoms Checklist–25** | | | |
| Depression | 291 | 15 | 0.87 |
| Anxiety | 295 | 10 | 0.82 |

*Note:* Results are from proxy reports by caregivers. Cronbach's alpha threshholds of α ≥ 90 (excellent), 0.7 ≤ α < 0.9 (good); 0.6 ≤ α < 0.7 (acceptable); 0.5 ≤ α < 0.6 (poor) and α < 0.5 (unacceptable).

(MD = −0.33; 95% CI: −0.58, −0.09). A similar trend was observed at 6- and 12-month follow-up. Except at 6 months (MD = −0.25; 95% CI: −0.46, −0.03), there was no difference between caregivers with moderate versus high levels of depressive symptoms rating for ASD risk in their respective children. A large statistically significant reduced likelihood for FI was reported among caregivers with the least depressive symptoms at baseline and 6 months but was of moderate statistical significance at 12 months (MD = −0.31; 95% CI: −0.57, −0. 06). Caregivers with moderate depressive symptoms reported a small but significantly reduced likelihood of FI in their respective children at baseline (MD = −0.23; 95% CI: −0.43, −0.03) and 6 months (MD = − 0.28; 95% CI: −0.49, −0.07) (Table 3).

## Discussion

We investigated the relationship between caregiver depression symptom level and developmental disorder risk scores in their dependent children from Uganda over the course of 1 year. Caregivers with the highest levels of depression had a similar proportion of CPHIV and CHUU with most of them being CHEU. Conversely, among caregivers with the least depressive symptoms, most of their dependent children were CPHIV with the least being CHEU. This may suggest that highly depressed caregivers experience stress due to the condition of their HIV-exposed, uninfected children. We found that caregivers with low levels of depression symptomology rated their dependent children as having a moderately lower likelihood for developmental disorder risk (i.e., ASD, ADHD and EBD), and functional impairment than those with moderate or severe range self-reported depression scores. Using thresholds of <0.33, 0.33 ≤ |MD| < 0.50 and |MD| ≥ 0.50 (corresponding to small, moderate or large clinical significance, respectively), low caregiver depression was thus a moderate clinically important predictor of lowered likelihood risk for ASD, ADHD, EBD and FI in dependent children per respective caregiver ratings. When conducting neuro-developmental disorder risk assessments and diagnoses in children, diagnosing professionals need to contextualize proxy reports of symptoms. This is because caregivers with- and without-depression may underreport or overreport symptoms in their dependent children, respectively.

Our findings are consistent with reports from a large cohort survey study of Taiwanese children of parents with versus without major

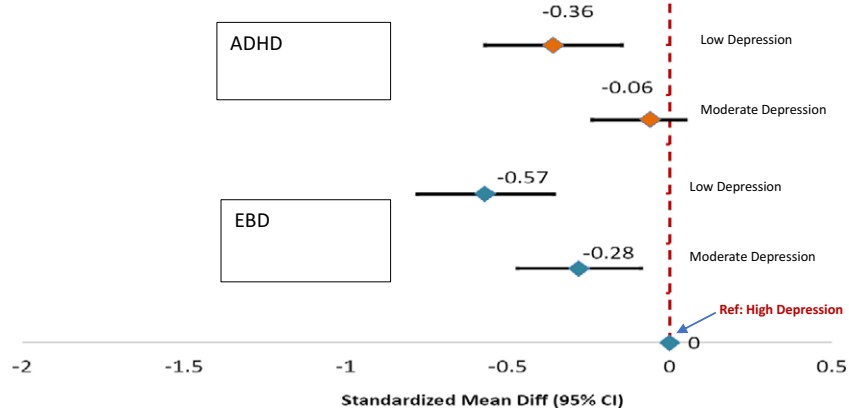

**Figure 1.** Association between caregiver depressive symptom level, attention deficit hyperactivity disorder (ADHD) and emotional behavioral disorder (EBD) in dependent children. *Note:* Results derived from multivariable regression model with adjustment for time, child (sex, age), caregiver (HIV-status, sex, age, education, lifetime adverse experiences, history of smoking and alcohol). Negative values show that caregivers risk perception for ADHD and EBD in their dependent children decreases with depressive symptoms. The reference group is caregivers with highest level of depression.)

**Table 3.** The relationship between caregiver depression symptomology level, autism spectrum disorder and functional impairment risk among children and adolescents in Uganda across time

| Outcomes | Depression level | Baseline Adjusted MD (95% CI) | 6 months Adjusted MD (95% CI) | 12 months Adjusted MD (95% CI) | Time * depression interaction |
|---|---|---|---|---|---|
| **Autism spectrum disorder** | | | | | **0.07** |
| | Low (n = 143) | **−0.33 (−0.58, −0.09)** | −0.35 (−0.60, −0.10) | −0.32 (−0.56, −0.08) | |
| | Moderate (n = 305) | −0.13 (−0.34, 0.08) | **−0.25 (−0.46, −0.03)** | −0.04 (−0.25, 0.16) | |
| | High (n = 155) | **Ref** | **Ref** | **Ref** | |
| **Functional impairment** | | | | | **0.02** |
| | Low (n = 143) | **−0.56 (−0.81, −0.31)** | **−0.51 (−0.77, −0.26)** | **−0.31 (−0.57, −0.06)** | |
| | Moderate (n = 305) | **−0.23 (−0.43, −0.03)** | **−0.28 (−0.49, −0.07)** | −0.01 (−0.22, 0.20) | |
| | High (n = 155) | **Ref** | **Ref** | **Ref** | |

*Note:* *Results include follow-up interval specific associations derived from multivariable regression models adjusted for dependent child's HIV status and age; caregiver sex, age, education, lifetime adverse experiences, history of smoking and alcohol use. MD = mean difference. Bold numbers represent statistically significant associations.

depression where it was found that compared to the former, children whose parents had major depression had a high likelihood to be diagnosed with ASD and ADHD (Lin et al., 2024). Further, similar elevated rates of behavioral problems in dependent children have been reported by American (Gartstein et al., 2009), Finnish (Liskola et al., 2021) and German (Müller et al., 2011) caregivers with respect to internalizing and externalizing behaviors – integral sub-domains often assessed for a diagnosis of developmental disorders. Similarly, Familiar and colleagues (Familiar et al., 2020; I. Familiar et al., 2016) found that caregivers from Zimbabwe, Uganda, South Africa and Malawi with high depressive symptoms reported more executive function problems in their 7- to 11-year-old children. Furthermore, among rural caregivers in Kenya (Laurenzi et al., 2021) on average depressed mothers rated their children as having more behavioral challenges. Our study findings support earlier observations and highlight that cognitive and mental health outcomes in dependent children, especially when reported by proxies, are likely influenced by the proxy respondent's affect.

There are mixed research findings pertaining to how maternal depression level impacts their behavioral ratings of dependent children. Some studies have reported that depressed caregivers are more accurate (i.e., they do not overreport or underreport) in behavioral ratings of their children stemming from their awareness of- and sensitization to mental health symptoms (Gartstein et al., 2009; Müller et al., 2011; De Los Reyes et al., 2015). Although our findings concur with the former (i.e., depression-distortion hypothesis), it may be important to consider the possibility that depressed parents could be more accurate in their rating of behavioral challenges (especially for externalized behaviors) of their children. The mixed findings suggest the need for triangulation of information from different cross informants before a clinical diagnosis is made (Liskola et al., 2021). Several triangulation studies, however, report discrepancies in parental ratings of behavioral manifestations especially when compared to teachers, therapists or their own respective dependent children's rating of themselves, (Hughes and Gullone, 2010; Liskola et al., 2021) for internalizing versus externalizing behaviors (Liskola et al., 2021). There is more congruence in assessments related to externalized versus internalized behaviors by raters (Müller et al., 2011; De Los Reyes et al., 2015; Liskola et al., 2021).

This incongruence has been attributed to the fact that most externalizing behaviors are overt (e.g., aggression, hyperactivity and antisocial behaviors). The converse is reported for internalizing behaviors like anxiety, being withdrawn or depressed, which are covert and harder to observe hence the need for alternative professional opinions before a clinical decision is made.

Unlike for adolescents, younger children are rated exclusively by their caregivers, which makes certain forms of assessment triangulation difficult. Gartstein et al. (2009), for example, found that compared to their respective sons and daughters, American mothers overreported externalizing and internalizing behavioral problems in their sons and daughters aged 10–14 years, respectively. A similar finding was reported among mothers in Germany who rated their children higher for internalizing and externalizing behavioral problems than teachers and therapists (Müller et al., 2011). The need for child and adolescent evaluations by different stakeholders, including their self-evaluations to counter potential bias from the different contexts, cannot be overstated. As found in this study, caregiver rating of their respective dependent children improved across time. It may, therefore, be important to defer diagnostic decisions if a caregiver is found to display traits of depression at the time of their respective children's assessment for developmental disorders.

Our study has limitations that should be considered in the interpretation of our findings. Specifically, information on depression was collected by caregiver reports during face-to-face interviews in our clinic. Such reports are susceptible to the influence of caregiver social desirability and differences in their recollection of specific child behaviors. These could have led to overreporting or underreporting of behavioral manifestations in their children and yet we did not have cross informants to counteract such potential bias. While caregiver baseline depression is treated as a predictor in relation to outcomes measured over time, it is indeed possible that the child's condition is fueling depression level of caregivers at baseline. Indeed, the bidirectionality of association is possible and cannot be excluded. Our study had the strength of participants being familiar with the assessment setting and process of being asked about their children's developmental history or functioning. They thus felt safe with subsequent ratings of their children. Further, our study included the use of longitudinal design with

repeated measures of developmental disorder risk in order to confirm findings or highlight changes over time. Additional strengths include a large sample size and the implementation of a rigorous analytic strategy controlling for multiple confounders. Our team has previously demonstrated the reliability of the BASC-3 (Zalwango et al., 2016), although for different outcomes. We have now established the reliability of the BASC-3 items assessing developmental disorder indices, functional impairment and RIs as well as subscales of the HSCL-25. It is also important to note that the HSCL-25 has been validated to assess depression in Uganda (Bolton et al., 2003; Ovuga et al., 2008; Tsai et al., 2012), although further validation (especially correlational studies) with gold-standard developmental disorder diagnosis and screening tools, is needed.

Our findings have ramifications for LMICs – settings (Herba et al., 2016) with a high patient-to-practitioner ratio as well as high rates of maternal depression – (Kamenov et al., 2017). Without a deliberate effort to understand and adjust for the potential influence of caregiver depression in child assessment for developmental disorders, there is an increased likelihood of misdiagnosis. This places pressure on already scarce public health resources, making it more likely that those in most need miss out, disadvantaging their ability to thrive in adulthood. This points to the need to support the primary caregivers' mental health, for example, through counseling, if we are to make a change for the child. Practitioners must be sensitized to the depression-distortion hypothesis and how to control for it in clinical and research settings. Also, future research needs to focus on the complimentary use of both caregiver and child self-rating for developmental disorder risk in LMICs. Information from adolescents' self-rating, performance-based psychological assessments and cross-informant and clinician ratings must be considered during clinical assessment. Routine mental health screening for caregivers should be conducted, particularly during pediatric developmental evaluations in similar settings. Depression affects women more than men and yet women contribute most to childcare in LMICs highlighting the need for caregiver-focused intervention strategies to enhance child wellbeing. Psychotherapy and pharmacotherapy have been shown to be reliable interventions for caregiver depression or stress level as well as leading to better functioning and quality of life (Cuijpers et al., 2020; Leichsenring et al., 2022; Cuijpers et al., 2023). More studies targeting depressed caregivers in clinical assessments are warranted to further understand the relationship between caregiver depression level and developmental disorder risk in their respective children (Herba et al., 2016). These studies could also take advantage of psychological interventions for parents and comparison of reporting patterns over time as treatment continues.

## Conclusion

The level of depression in caregivers is associated with how they evaluate the behavior of their dependent children. We highlight the need to consider parental depression when dealing with parental reports about child behavior and psychopathology in both research and clinical settings. Whenever possible, multidimensional assessments of developmental disorders and behavioral outcomes should be used, including, as appropriate the child's, the caregiver's or other respondent's perspective.

**Open peer review.** To view the open peer review materials for this article, please visit http://doi.org/10.1017/gmh.2025.10078.

**Supplementary material.** The supplementary material for this article can be found at http://doi.org/10.1017/gmh.2025.10078.

**Data availability statement.** Data supporting findings herein will be availed by the corresponding author (JEA), upon reasonable request and subject to data sharing agreements.

**Acknowledgments.** The authors would like to thank the study participants and Kawaala Health Center IV staff. Equally, the authors extend their gratitude to the field research staff Nakigudde Gorreth, Esther Nakayenga, Faridah Nakatya, Irene Asiingura, Arnold Katta, Phiona Nalubowa and Isabella Achokotho Akol (Administrative Support).

**Author contribution.** Conceptualization: J.E.A. and A.E.E.; formal analysis, J.E.A, A.E.E. and A.S.; writing—original draft preparation; J.E.A., A.E.E., A.S. and B.G.; writing—review and editing, J.E.A., A.E.E., A.S., A.C., S.Z. and B.G.; project administration, S.Z. All authors have read and approved the final version of the manuscript.

**Financial support.** Research funding for data collection leading up to this work was provided by the National Institutes of Health (grant numbers 3R21HD088169-02S1 and R21HD088169; and the CIPHER Grant program of the International AIDS Society (grant number 327-EZE).

**Competing interests.** The authors declare none.

**Ethics statement.** This research was conducted with strict adherence to the Declaration of Helsinki. The study was approved by the Institutional Review Boards (or Ethics Committee) of Michigan State University (IRB Protocol numbers: 16-828 and 205), Makerere University College of Health Sciences, School of Medicine (Protocol REC REF numbers: 2017-017 and 2018-099) and the Uganda National Council for Science and Technology (Protocol numbers: SS4378 and HS 2466).

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
