## [Reviewer Report]

Thank you for the opportunity to review ‘Association of Adult Caregiver Depression with Developmental Disorder Likelihood in Ugandan Children Perinatally Exposed and Unexposed to HIV’. The topic is of potential interest for the audience of the journal. The manuscript has a few shortcomings and overall requires proof reading to address grammar and style. Other comments by section are as follows.

Abstract

Introduce acronym when first used (FI)

Ln 51 is not clear and needs rephrasing

Introduction

Given that the sample for this study is based on HIV exposure, and HIV is mentioned in the title, this section should be more tailored to research focusing on children living and/or exposed to HIV and their caregivers.

Ln 119 cites studies as ‘Boivin and colleagues’ yet the references provided are from articles by Familiar. Please revise

Ln 124 Phrasing is hard to understand. Please revise

Methods

Provide the breakdown for CPHIV, CHEU and CHUU in line 141

Please clarify when were CHEU and CHUU recruited (eg. with the 2017 or 2021 studies or independently?)

Ln 160 Which is the parent study? The section above references 2 studies from where CPHIV were recruited, in addition to the recruiting methods used for CHEU and CHUU. Did these criteria apply to all?

Ln 163 What is the study area? Only attendance to the Kawaala clinic is mentioned

The HSCL also measures anxiety, yet results are only presented for depression. Was there a specific reason why? Seems like a lost opportunity to add results

Similarly, given that 3 HIV exposure groups were included and that there were differences in the level of caregiver depression within each group, why was HIV status only adjusted for?

Results

Ln 239. Study design characteristics should be reported in the methods section

Ln 248 Not clear what this sentence says, please revise

Discussion

Ln 294 Phrase is repetitive and unclear. Please revise

Although the depression-distortion hypothesis and the sensitization to mental health symptoms are discussed, the interpretation of results is assuming unidirectionality of the relationship between caregiver depression and child ratings. However, a more complete analysis should also take into account the possibility of bidirectionality, in that depression symptoms may be due to developmental challenges in the dependent child.

---

## [Reviewer Report]

This manuscript investigates whether caregiver depressive symptoms are associated with increased perceived risk of developmental disorders in their children. The study is based on longitudinal data collected in Uganda and uses multivariable models to evaluate changes over a 12-month follow-up.

The central hypothesis is clearly stated by the authors:

“We hypothesized that caregivers with more symptoms of depression would report greater risk for their child to have a developmental disorder, consistent with the depression–distortion hypothesis.”

This is a relevant research question, particularly in low-resource settings where caregiver-reported data often play a key role in early identification of developmental concerns.

The manuscript is well organized and presents findings clearly. However, there are several areas that would benefit from clarification or improvement, particularly related to methodology, statistical reporting, justification of setting, and table/figure presentation.

Specific comments:

1. Study design and methodology

• The study is a secondary analysis of a prospective cohort. While this is stated in the Methods section, it should also be mentioned clearly in the Introduction.

• There is no mention of a sample size calculation or power analysis. While not always required for secondary analyses, it would be helpful to include a brief discussion on whether the available sample size was adequate to detect group differences across depression strata.

• Sampling was non-random and based on clinic recruitment and social networks. This limits generalizability and should be noted explicitly as a potential source of selection bias.

• The instruments used (BASC-3 and HSCL-25) were translated into Luganda. However, there is no evidence provided of formal psychometric validation in the local population. This should be acknowledged as a limitation.

2. Table 1

• Variables were compared across depression groups using one-way ANOVA (continuous) and chi-square tests (categorical). The label “ANOVA/χ²” should be revised to specify: “One-way ANOVA (continuous variables) and chi-square test (categorical variables).”

3. Table S1

• This table is useful for observing unadjusted trends in standardized scores over time. However, sample sizes (n) are only reported for ASD and should be included for the remaining domains (ADHD, EBD, FI) for consistency.

• The table does not specify whether the values are unadjusted means. This should be clarified.

4. Table 2

• The column labeled “M (95% CI)” should be renamed to “Adjusted MD (95% CI)” to clearly indicate that the values represent adjusted mean differences rather than raw means.

• Sample sizes for each depression category (Low, Moderate, High) are missing. Including the n per group—either within the table or as a footnote, would allow readers to assess the robustness of the estimates.

• The “Time × Depression” interaction column reports only p-values. Reporting the corresponding interaction coefficients and confidence intervals would provide a more complete understanding of the longitudinal association.

• Although the statistical analysis section refers to Cohen’s d thresholds, the clinical interpretation of the MD values is not addressed. A concise statement on whether the differences correspond to small, moderate, or large effects would strengthen the interpretation of results.

5. Figure 1

• Including sample sizes per group and exact numerical values for the point estimates and confidence intervals, either within the figure or as a supplementary table, would improve interpretability.

• The reference group used for comparisons (i.e., caregivers with high depressive symptoms) should be clearly stated in the figure caption to avoid ambiguity.

• The direction of negative values is not explained. It would be helpful to state explicitly that negative values represent lower standardized scores compared to the reference group.

• Acronyms in the title (ADHD, EBD) should be defined at first mention, either in the title or in the figure legend.

6. Discussion

• The interpretation of results is consistent with the stated hypothesis: “caregiver depression level would be associated with higher risk perception and thus higher developmental disorder risk scores in their children.” However, the discussion omits alternative explanations. For greater analytical depth, the authors should briefly mention evidence showing that caregiver depression does not always result in risk overestimation, e.g., studies indicating accurate or even underreported symptoms by depressed caregivers.

• Policy implications are presented in general terms. The discussion should include specific and actionable recommendations, such as incorporating routine caregiver mental health screening as part of pediatric developmental assessments in similar settings.

7. Justification of setting

• The study is conducted in Uganda, but this is not clearly justified in the Introduction.

• Since the setting is central to the relevance of the findings (e.g., high burden of HIV, limited mental health infrastructure), a brief explanation should be included early on to clarify why this context is particularly suitable for the research question.

---

## [Reviewer Report]

Thank you very much for the opportunity to review this manuscript. The paper is well written, and the authors address a gap in mental health and developmental disorders research by examining the relationship between caregiver mental health and developmental disorders rating outcomes in Uganda. The study, which employed a 12-month longitudinal design, investigated the hypothesis that a caregiver’s depression level is associated with their subjective ratings of their dependent children’s risk for developmental disorders. The study’s central finding is that caregivers with higher depressive symptoms consistently rated their children as having a greater likelihood of developmental disorders, including autism, ADHD, suggesting that caregiver mental health is a crucial factor to consider during child assessments.

Below are some suggestions for the author’s consideration.

Abstract

The paper included populations exposed, infected or uninfected with HIV. Could you please mention this in the abstract, as you point this out in the title of the paper as well?

Introduction

Line 69: The prevalence increased by 48.1% to what percentage? Could you please add this?

Lines 69, 72, 73: Could you please add the proper in-text citation for the GBD report?

Lines 126-133: When stating the objective of the study in the introduction, you don’t mention the population included. Could you please add that?

Methods

Line 207: Socioeconomic status was defined as the presence or absence of an income source. Would it be fairer to say that this question is a proxy for SES instead of a definite measure of SES.

Line 208 – 209: Information on life adversity and living with a sexual partner was gathered using a structured questionnaire. Could you please share this questionnaire as an appendix so that we know the questions asked?

I also notice that the number of adverse life experiences wasn’t a covariate adjusted for in the models. Is there a reason this wasn’t included? Considering that exposure to adverse life events is associated with depressive symptoms? The authors do find that ‘caregivers had a significant difference in the number of adverse lived experiences, with caregivers in the high depression symptomology category having experienced the most adversity.’

Results

Line 248 to 249 - Low vs. high depression symptom caregivers rated their children as moderately lower but significant for all developmental disorders and FI (all p < 0.05) at 12 months

Could you please phrase this sentence differently? It is hard to understand as it is.

Discussion

CHUU and CHIV have almost a similar amount of moderation and high depression from the observations in Table 1. Could you please discuss this in some detail?

Limitations

The main tools used (HSCL, BASC) have not been psychometrically explored in this setting, meaning there is missing evidence on validity and reliability. You mention this in the methods section. Could you please add this to the limitations section as well? While using the sample to get standardised scores and summation are pragmatic approaches, there is a possibility that the approaches may not be psychometrically sound in this cultural context.

References

Please check the reference list, line 446, the title of the paper is incomplete.

---

## [Reviewer Report]

Authors have addressed all comments raised and significantly revised the manuscript.

I have no further comments

---

## [Reviewer Report]

The revised manuscript shows clearer organization and stronger methodological transparency in response to prior reviewer feedback. Key aspects such as sample recruitment, the use of SES as a proxy, covariate adjustments, and psychometric considerations have been more clearly explained.

Tables and figures are now more readable, with added clarity regarding group sizes, statistical labels, and reference categories. The discussion includes expanded analysis on possible bidirectional effects and variation in caregiver reporting, which strengthens the interpretation of the findings.

No further methodological concerns remain from my side.

---

## [Reviewer Report]

Thank you to the authors for the careful attention to the feedback and comments. I do not further comments and wish them the best with the next steps.